# The Impact of Highly Effective Treatment in Pediatric-Onset Multiple Sclerosis: A Case Series

**DOI:** 10.3390/children9111698

**Published:** 2022-11-05

**Authors:** Paolo Immovilli, Paola De Mitri, Veronica Bazzurri, Stefano Vollaro, Nicola Morelli, Giacomo Biasucci, Fabiola Magnifico, Elena Marchesi, Maria Lara Lombardelli, Lorenza Gelati, Donata Guidetti

**Affiliations:** 1The Neurology Unit, Emergency Department, Guglielmo da Saliceto Hospital, Via Taverna 39, 29121 Piacenza, Italy; 2The Pediatric Unit, Maternal and Child Department, Guglielmo da Saliceto Hospital, Via Taverna 39, 29121 Piacenza, Italy

**Keywords:** pediatric onset multiple sclerosis, POMS, highly effective treatments, disease-modifying drugs, DMD

## Abstract

Introduction: Pediatric-onset multiple sclerosis (POMS) is characterized by high inflammatory disease activity. Our aim was to describe the treatment sequencing and report the impact highly effective disease-modifying treatment (HET) had on disease activity. Materials and Methods: Five consecutive patients with POMS were administered HET following lower efficacy drug or as initial therapy. Data on treatment sequencing, relapses and MRIs were collected during the follow-up. Results: Our patients had an average age of 13.8 years (range 9–17) at diagnosis and 13.4 years (range 9–16) at disease onset, and 2/5 (40%) POMS were female. The pre-treatment average annualized relapse rate was 1.6 (range 0.8–2.8), and the average follow-up length was 5 years (range 3–7). A total of 2/5 (40%) patients were stable on HET at initial therapy, and 3/5 (60%) required an escalation to more aggressive treatment, even if two of them had been put on HET as initial treatment. Four out of five patients (80%) had No Evidence of Disease Activity-3 status (NEDA-3) at an average follow-up of 3 years (range 2–5). Conclusion: It has been observed that in a recent time period all the cases had prompt diagnosis, early HET or escalation to HET with a good outcome in 80% of the cases.

## 1. Introduction

Pediatric-onset multiple sclerosis (POMS) is generally defined as multiple sclerosis (MS) with an onset before the age of 18 and accounts for about 3 to 5% of all MS cases [1,2]. It is characterized by active inflammation, a high rate of disabling relapses, followed by optimal remissions [3]. Children have a lower disability risk than adult-onset multiple sclerosis (AOMS) within the first 10 years from diagnosis, a longer time lapse to the secondary progressive phase but reach disability milestones younger [4]. Furthermore, the characteristic cognitive symptoms, i.e., fatigue and depression, place a high burden on these patients, strongly impacting their quality of life and education [4].

A recent study of POMS reported a reduction in the risk of persistent disability by 50% to 70% in a recent “diagnosis epoch”, probably due to therapeutic and managing standards improvement [5].

Highly effective disease-modifying treatments (HETs) have been reported to reduce the annualized relapse rate by 50% or more [6]. Other authors have stated that early treatment of MS is efficacious in preventing disability accrual [7], as is the early introduction of HET [8], even as first therapy [9]. Fingolimod (HET) and teriflunomide (moderate-efficacy treatment) have been approved by the European Medicines Agency (EMA) for POMS and observational studies have reported administering almost all the other MS disease-modifying drugs (DMDs) adopted for adults, also in POMS. However, the paucity of randomized clinical trials (RCTs) leaves the debate open. 

*Aim*: to provide data on the management of POMS by describing the treatment sequencing to second-line therapies in five consecutive patients with POMS and report the impact HET had on disease activity.

## 2. Materials and Methods

This retrospective case series includes all the consecutive patients with POMS treated at the Multiple Sclerosis Centre, Piacenza, Italy, between January 2015 and March 2022. The MS diagnosis was made on the basis of the McDonald criteria 2010 revision [10] until 2018 and then the McDonald 2017 revision [11] from 2018 on. 

The effectiveness of DMDs was assessed by no evidence of disease activity-3 status (NEDA-3): NEDA-3 status is defined as the absence of relapses, disability progression and new/enlarging or enhancing lesions on the brain and cervical spinal MRI [12]. HET was defined as DMDs that reach more than 50% reduction in the relapse rate, according to Samjoo et al. [6] (Table 1).

Cognitive functions have been evaluated through an extensive neuropsychological battery that assessed memory, language, attention, executive functioning and intelligence quotient (IQ) at the moment of diagnosis. We used the Symbol Digit Modality Test (SDMT) for the follow-up. 

Ethical committee approval was waived by the local review board due to the study design (case series). Written informed consent was obtained from the patients’ parents, and all the data were stored in anonymized datasets. 

## 3. Results

Our patients had an average age of 13.8 years (range 9–17) at diagnosis, 13.4 years (range 9–16) at onset and 14.4 years (range 13–17) at first DMD; 2/5 (40%) were female and 3/5 (60%) male. The pre-treatment average annualized relapse rate was 1.6 (range 0.8–2.8), the average follow-up length was 5 years (range 3–7) and the average last EDSS was 1 (range 0–2.5).

A total of 2/5 (40%) patients were stable on HET at initial therapy, and 3/5 (60%) required an escalation to more aggressive treatment even if two of them had been put on HET as initial treatment. Four out of five patients (80%) had a NEDA-3 status at an average follow-up of 3 years (range 2–5). The two cases treated by HET as first therapy had a NEDA-3 status at three-year follow-up. Table 2 illustrates the main clinical characteristics of all the cases described.

Cognitive functions have been evaluated through an extensive neuropsychological battery: we did not find any cognitive impairment at the moment of diagnosis. We used SDMT for the follow-up and it was unremarkable every year.

### 3.1. Case 1

A 13-year-old girl was admitted with an incomplete internuclear ophthalmoplegia (INO) and an expanded disability status scale (EDSS) score of 2.0. Magnetic resonance imaging (MRI) evidenced a pontine T2-FLAIR hyperintense lesion, five periventricular and two cervical spine lesions at C1 and C6. A cerebrospinal fluid (CSF) examination evidenced 12 intrathecal oligoclonal bands. Steroid pulse therapy (SPT) was administered and early, complete remission was reached within 1 month. A diagnosis of a clinically isolated syndrome (CIS) was made and a weekly therapy of intramuscular interferon β-1a (at 30 μg once a week) was started.

However, 10 months later she was re-admitted with mild right retrobulbar optic neuritis (ON) and an EDSS score of 1. SPT improved vision within 3 days and a follow-up MRI did not evidence any new or active lesions. The patient had a mild left-sided sensory and motor relapse 12 months later and an EDSS score of 1.5. Once again, SPT led to quick remission, although the follow-up MRI evidenced one new cerebral enhancing and two cervical spine lesions in the left dorsal column segments C5 and C6. As interferon β-1a treatment was not effective, it was discontinued.

A double ELISA serological test for the JC virus (JCV Stratify Test) [13] was low titer positive (index 0.43), indicating a low risk of progressive multifocal leukoencephalopathy (PML) development and intravenous natalizumab (at 300 mg every 28 days) was started. However, as the patient developed angioedema and a diffuse rash at the second infusion, she was switched to fingolimod at the normal adult oral dose of 0.5 mg/day.

One month later, she complained of a minor sensory relapse, whilst her EDSS score remained stable. Brain and cervical MRI scans were performed 7 months later (“rebaseline” MRI) for use as a new baseline for future comparisons and evidenced a new 6 mm peritrigonal lesion.

At the 7-year follow-up, the patient’s EDSS is still stable at 1.0, and she has had a NEDA-3 status for 5 years, i.e., since she started fingolimod treatment.

### 3.2. Case 2

A 9-year-old boy was admitted with diplopia and INO. An MRI evidenced a large middle pontine demyelinating lesion and three supratentorial lesions. The CSF examination evidenced nine intrathecal oligoclonal bands, whilst anti-MOG and anti-AQP4 antibodies were negative. SPT was administered and led to complete remission. Follow-up MRI did not evidence any new, enlarging or active lesions.

One year later, he complained of urinary urgency and of being unable to run properly, but his gait was normal. During the relapse, his EDSS score was 2.0, and the brain and spinal MRI evidenced improvement of the old lesions. The diagnosis of POMS was made on the basis of two relapses (dissemination in time) in two different functional system (dissemination in space), according to the revised McDonald criteria [10]. SPT led to a slow, but complete remission, where his EDSS score had returned to zero. Brain and spinal cord MRI was unchanged after 3-month, 6-month and then yearly follow-ups up to 3 years.

However, he had a major relapse at the age of 13, with ataxic paraparesis and INO; his EDSS score was 7.0 and an MRI evidenced a new large demyelinating pontine lesion. SPT and natalizumab were started, which led to complete remission after rehabilitation and an EDSS score of 1.0.

Although he had three minor relapses between 13 and 15 years of age, SPT led to complete remission and MRI evidenced no new lesions.

As he had three relapses in 18 months whilst on natalizumab, he was switched to alemtuzumab, but 4 months later he had another major relapse and an EDSS score of 5.0. After SPT, he recovered completely within 2 weeks. Neither his brain nor spinal cord MRI evidenced any new, enlarging or enhancing lesions. At the time of writing, he is 16 and has had his second alemtuzumab cycle. He has had no relapses in the last 7 months, and his EDSS score is currently 1.5.

### 3.3. Case 3

A 14-year-old girl was admitted with a bilateral ON and a chiasmatic involvement. A week later, she had a spinal cord syndrome with mild paraparesis and an EDSS score of 3.5. Her anti-MOG and anti-AQP4 antibodies were negative, whilst multiple demyelinating lesions, with a high lesion load, were observed on her brain and spinal MRIs and the CSF examination evidenced 11 intrathecal oligoclonal bands. SPT led to a quick complete remission and a diagnosis of POMS. One month later, she had a brainstem syndrome, symptomatic for vomiting, ataxia and dizziness, an EDSS score of 7.0 and her MRI evidenced three new spinal Gadolinium-enhancing lesions. Natalizumab was started, though her JCV stratify test was positive and had a high titer (high PML risk) [13]. Six months later, she was switched to an extended interval dose (EID) every 6 weeks, and 3-monthly follow-up MRIs were scheduled to check for any early radiological signs of PML. At the 3-year follow-up, her EDSS score was zero, and at the time of writing she has a NEDA-3 status.

### 3.4. Case 4

A 15-year-old boy was admitted with ataxia, diplopia and vomiting. He reported having had neurological symptoms twice in the previous year, characterized by bilateral paresthesias in his legs, of 1 month duration. The neurological examination revealed bilateral nystagmus and an ataxic gait, and his EDSS score was 5.5. The CSF analysis evidenced the presence of several identical oligoclonal bands in both plasma and CSF and an isolated monoclonal band in the CSF. SPT led to complete remission. A follow-up MRI at 2 months detected a new brainstem lesion, the JCV stratify test was positive with a low titer and natalizumab was started.

As it was March of 2020, little data was available on a severe COVID-19 risk during MS DMDs. Therefore, as we were somewhat concerned about the risks linked to fingolimod lymphopenia, it was decided to start him on natalizumab. The three-year neurological follow-up was unremarkable, and at the time of writing, he has a NEDA-3 status.

### 3.5. Case 5

A 17-year-old boy was admitted with diplopia. He reported having had two episodes of paresthesias in his legs in the past, one when he was 15 and another at 16. A neurological examination revealed a mild right eye adduction deficit, and his EDSS score was 2.0. The CSF examination showed the presence of three intrathecal oligoclonal bands, and MRI evidenced a significant brain lesion load with several black holes (i.e., a T1-weighted hypointense lesions) and two spinal lesions. SPT led to complete remission. Two months after discharge, he had a mild relapse with an EDSS of 2.0. SPT led to complete remission.

He was started on fingolimod but had two relapses during the first year of therapy. The 6-month follow-up MRI evidenced two new lesions, and the 12-month MRI another new lesion. Consequently, he was switched to natalizumab. However, two relapses occurred within the first 7 months, and the 7-month follow-up MRI evidenced four new lesions, one in the brain and three in the spine. Natalizumab was substituted by alemtuzumab due to the aggressive disease course. At the time of writing, after 2 years and 8 months of alemtuzumab, he has had no relapses, his EDSS is stable at 2.5 and the follow-up MRIs have not evidenced any new lesions; currently, he has a NEDA-3 status.

## 4. Discussion

POMS is a relatively rare disease and diagnosis may be quite challenging. The 2017 revision of the McDonald criteria [11] states that MS diagnostic criteria could be used in children above 11. The fact that there are few RCTs on POMS in the literature and a paucity of evidence to guide treatment decisions can be justified by several factors, e.g., they are frequently underpowered, there are more long-term safety concerns and additional ethical issues in pediatric populations [14,15]. However, the PARADIGMS trial [16] led to the approval of fingolimod for POMS and the TERIKIDS trial [17] to teriflunomide approval in Europe but not in the USA. Furthermore, there are many observational studies on virtually all the DMDs used for AOMS, and some RCTs, aimed at providing data to guide treatment decisions in POMS, are still ongoing.

Noteworthy is the fact that a recent large propensity score-matched observational cohort study on 741 POMS [1] reported that HET is probably more effective than older agents in preventing relapses and slowing down radiologic disease progression in POMS.

The diagnosis of clinically isolated syndrome was made at disease on-set on the basis of the 2010 McDonald criteria [10] in case 1, and a moderate efficacy DMD was started. One year later, a relapse occurred, and the patient was switched to HET with a good outcome.

As our case 2 was in 2015, the diagnosis was made on basis of the 2010 McDonald criteria [10], and at disease onset, a diagnosis of a clinically isolated syndrome was made. As there was only clinical activity without radiological activity, a diagnosis of POMS was made on a clinical basis. However, there was a significant delay between disease onset, POMS diagnosis and his being switched to HET.

Conversely, the diagnoses of POMS in cases 3 and 4 were made at clinical onset, based on the new 2017 McDonald criteria [11], and HET was started, leading to a NEDA-3 status at three years.

Case 5 was started on HET and escalated to alemtuzumab within two years from POMS diagnosis, which was made according to the McDonald 2017 revised criteria [11].

Baroncini et al. [5] described a large cohort of POMS patients divided into different “*diagnosis epoch*” (before 1993, from 1993 to 1999, from 2000 to 2006 and from 2007 to 2013) and reported a reduction in the risk of persistent disability during the later period. They attributed this important finding to a higher number of patients treated with DMDs, especially high-potency drugs, more recently. The study was carried out on a large POMS population, with an average age of 15.2 years at onset and 22.1 years at diagnosis. A total of 22% of patients started DMDs during pediatric age. However, from 2007 to 2013, this rose to 39%.

Our data collection was based on the use of the McDonald criteria 2010 and 2018 revisions. All patients were diagnosed in pediatric age (average 13.8 years), and treatments, also HETs, were started within a few months after diagnosis in 4 out of 5 patients. Our findings are in line with those of Baroncini et al. [5] published in 2021.

Although our small case series showed a highly active disease course in all the patients, 80% had a NEDA-3 status during the first two years of HET: this observation is consistent with a greater effectiveness of newer therapies, as reported by Krysko et al. [1]. Two patients had relapses and new MRI lesions even during the first HET and were escalated to alemtuzumab because of its inductive immunosuppressive mechanism of action. Despite the paucity of literature data on the use of alemtuzumab in POMS [18,19] and the risk of autoimmunity, the risk of disability accrual if a sub-optimal treatment was continued outweighed the potential treatment adverse events.

Moreover, achieving a NEDA-3 status during the first two years of treatment has been reported to predict a 78% probability of no disability accrual at 7 years in AOMS [12] and NEDA-3 has been proposed as a reasonable treatment goal when clinicians have to treat highly active MS [20].

Fingolimod and natalizumab are the only HETs approved by the regulatory agency in Italy, and the use of off-label HETs (i.e., alemtuzumab, anti-CD20 and cladribine) should be reserved for non-responders to approved HETs. The PARADIGMS trial [16] reported that fingolimod reduced by 82% the adjusted annualized relapse rate in POMS, and in the REVEAL study [21], the annualized relapse rate was 90% lower with natalizumab compared to fingolimod. Moreover, fingolimod was not considered to be a treatment option for non-responders to natalizumab on the basis of the REVEAL study results [21].

We used natalizumab as the first DMD in cases 2, 3 and 4, because we chose the more effective drug in the presence of multiple negative prognostic factors (e.g., spinal cord or brainstem relapse, high EDSS, high MRI lesion load). In cases 3 and 4, natalizumab was used, even if the stratify JCV test was positive: PML risk was managed with EID natalizumab in all the cases. Moreover, when the JCV stratify test was high titer positive (case 3), a continuous MRI follow-up was scheduled every 3 months, with the purpose being to detect early PML radiological signs. In the NOVA study [22], natalizumab EID showed similar efficacy with respect to standard dosing, so it is still considered HET. In the NOVA study [22], patients were switched to EID after one year of standard dosing. Case 4 switched to EID after one year of standard dosing, while case 3 switched after 6 months, and it is questionable if an early initiation of EID could be considered a HET. When natalizumab was not effective, we switched to alemtuzumab, and when it was not tolerated, we switched to fingolimod.

HETs have important safety concerns, but in our small case series, fingolimod was well tolerated and we did not observe any adverse event. A patient switched from natalizumab to fingolimod due to angioedema and skin rash, and patients treated with alemtuzumab complained of infusion-related reactions, but no serious adverse event has been observed.

We are aware that our study does have some limitations, e.g., the small number of cases, which does not allow for inferential statistical analysis, the absence of patients treated with anti-CD20 and cladribine, which does not allow for the generalizability of the observations to all HETs, and the DMD classification, which does not provide a categorization of HETs able to yield data on drug effectiveness.

The strength of our study lies in the fact that we were able to take advantage of the most recent diagnostic criteria (10–11) for early treatment, starting with HETs, that were later switched to more effective drugs, when required.

## 5. Conclusions

Hopefully, even if our case series is small, our contribution will help evidence gaps in literature and promote further research on the early introduction of HET in POMS.

## Figures and Tables

**Table 1 children-09-01698-t001:** Highly effective (HETs) and moderate-effective treatments (METs).

HETs	METs
Alemtuzumab	Injectables (e.g., interferons and glatiramer acetate)
Natalizumab	Dimethyl-fumarate
Anti-CD20 (e.g., ocrelizumab, ofatumumab)	Teriflunomide
Sphingosine-1-phospate modulators (e.g., fingolimod, Siponimod, ozanimod, ponesimod)	

**Table 2 children-09-01698-t002:** Clinical Characteristics.

Case	Case 1	Case 2	Case 3	Case 4	Case 5
Gender	Female	Male	Female	Male	Male
Age at onset (year-old)	13	9	14	14	16
Age at diagnosis (year-old)	13	10	14	15	17
Age at first DMD	13	13	14	15	17
ARR before treatment	0.8	1	2.8	1	2.3
First DMD (ARR on treatment)	Interferon (1.6)	Natalizumab (2)	Natalizumab (0)	Natalizumab (0)	Fingolimod (2)
Second DMD (ARR on treatment)	Natalizumab (0)	Aletuzumab (1.1)			Natalizumab (3.4)
Third DMD (ARR on treatment)	Fingolimod (0)				Alemtuzumab (0)
year since NEDA-3 status	5	0	3	2	2
Follow-up (years)	7	7	3	3	5
Last EDSS	1	1.5	0	0	2.5

DMD: disease-modifying drug; ARR: annualized relapse rate; NEDA-3: no evidence of disease activity-3; EDSS: expanded disability status scale.

## Data Availability

Researchers will provide the data upon reasonable request.

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
