# Peer review of "The Impact of Highly Effective Treatment in Pediatric-Onset Multiple Sclerosis: A Case Series"

_children, 2022, doi:10.3390/children9111698_

Round 1
Reviewer 1 Report
The authors presented a case series of five pediatric patients with MS who had been treated with high-efficacy disease‑modifying therapies (HET). The report provides real-world data regarding HET in pediatric MS. These data, though very preliminary, could be of some clinical value, as studies and experiences regarding HET in pediatric MS have been relatively limited. However, in my opinion, several issues are inadequately addressed in this paper. 1. "High-efficacy disease‑modifying therapies (HET)" are the focus of this paper. HET is, however, not clearly defined by the authors. Teriflunomide was mentioned by the authors (line 46), but it is generally viewed as moderate-efficacy rather than high-efficacy therapy. 2. Among drugs classified as HET, only 3 (fingolimod, natalizumab, alemtuzumab) were used in the patients reported here. It seems that treatment responses to different HET were different (for example, in case 2 and case 5). In case 5, stabilization of disease activity was achieved only after the third HET was used. I wonder if a conclusion that is generalizable to all HET could be made from this small case series. Perhaps the authors could discuss about their experiences with regard to individual HET (particularly natalizumab, which had ever been used in all five cases). 3. The rationale behind the choice of different HET should be better explained in some cases. For example, what was the consideration that natalizumab, rather than fingolimod, was chosen as the first HET in case 2, and case 3 (in the presence of high JC virus titer). 4. One of the concerns about HET, as compared to moderate-efficacy disease-modifying therapies, is their higher rate of adverse effects. I would suggest that the authors systematically describe and discuss about the side effect profiles of HET based on their experiences with these 5 patients. 5. The authors used NEDA-3 as evidence of treatment response. Did they also assess the cognitive performance of these patients? Cognition is included in NEDA-3 plus, and it is an even more important issue in pediatric MS. Minor: 1. line 17, "2/5 consecutive POMS were ...." Please correct or clarify. 2. in the third and fourth paragraphs of Discussion section, "Polman criteria" -> Please correct or clarify (I think it should be McDonald criteria). 3. Proofreading and editing of the manuscript may be necessary.Author Response
We thank the reviewers for their comments, which have allowed us to improve out paper.
We hope that it will now be deemed fit for publication in Children.
Please find below a point-by-point answer to the reviewer's comments.
Reviewer #1
Major Comments
- Comment #1 "High-efficacy disease‑modifying therapies (HET)" are the focus of this paper. HET is, however, not clearly defined by the authors. Teriflunomide was mentioned by the authors (line 46), but it is generally viewed as moderate-efficacy rather than high-efficacy therapy.
Answer: The definition of HET (He et al.) has been added in Materials and Methods. It has been clarified that teriflunomide is a moderate-efficacy therapy.
- Comment #2: Among drugs classified as HET, only 3 (fingolimod, natalizumab, alemtuzumab) were used in the patients reported here. It seems that treatment responses to different HET were different (for example, in case 2 and case 5). In case 5, stabilization of disease activity was achieved only after the third HET was used. I wonder if a conclusion that is generalizable to all HET could be made from this small case series. Perhaps the authors could discuss about their experiences with regard to individual HET (particularly natalizumab, which had ever been used in all five cases).
Answer: In the discussion, it has been added a paragraph describing our experience with the individual HET. Furthermore, in the study limitation paragraph, it has been discussed that the absence of children treated with anti-CD20 and cladribine does not allow for a full generalizability of our observations to all the HETs.
- Comment #3: The rationale behind the choice of different HET should be better explained in some cases. For example, what was the consideration that natalizumab, rather than fingolimod, was chosen as the first HET in case 2, and case 3 (in the presence of high JC virus titer).
Answer: In the discussion section, we better explained the rationale behind natalizumab treatment in JCV positive children and PML risk management.
- Comment #4: One of the concerns about HET, as compared to moderate-efficacy disease-modifying therapies, is their higher rate of adverse effects. I would suggest that the authors systematically describe and discuss about the side effect profiles of HET based on their experiences with these 5 patients
Answer: In the discussion section a paragraph has been added to discuss systematically HET adverse events in our case series.
- Comment #5: The authors used NEDA-3 as evidence of treatment response. Did they also assess the cognitive performance of these patients? Cognition is included in NEDA-3 plus, and it is an even more important issue in pediatric MS
Answer: A paragraph in the discussion has been added to explain the cognitive assessment.
Minor Comments
- Comment # 1: line 17, "2/5 consecutive POMS were ...." Please correct or clarify.
Answer: it has been corrected
- Comment #2: in the third and fourth paragraphs of discussion section, "Polman criteria" -> Please correct or clarify (I think it should be McDonald criteria).
Answer: Polman criteria has been corrected in 2010 McDonald criteria
- Comment #3: Proofreading and editing of the manuscript may be necessary.
Answer: Manuscript editing and proofreading has been done

Reviewer 2 Report
Thank you for this case series on highly effective disease modifying therapy in pediatric MS patients.
Could you please define what constitutes highly effective therapy? Maybe a table that lists the HET drugs compared to less effective therapy.
For case 2, line 115 - you state that the patient could not run but only had an EDSS of 2. Could you clarify if there was a gait/walking problem and how that would only lead to an EDSS of 2?
line 126 - it states that there were significant relapses but no new MRI lesions. Could you clarify if there were really no new MRI lesions at the time of the relapse?
line 153 - Please clarify the single intrathecal oligoclonal band and several bands in both plasma and CSF?
line 187 - you state that the 2017 criteria were the first to apply to children but the 2011 criteria did include children. The 2017 paper states that the McDonald criteria (including MRI and CSF) were most applicable to children age 11 and up.
Author Response
We thank the reviewers for their comments, which have allowed us to improve out paper.
We hope that it will now be deemed fit for publication in Children.
Please find below a point-by-point answer to the reviewer's comments.
Reviewer #2
- Comment #1: Could you please define what constitutes highly effective therapy? Maybe a table that lists the HET drugs compared to less effective therapy.
Answer: The definition of HET (He et al.) has been added in Materials and Methods. It has been added a table that lists MS highly-effective (HETs) and moderate-effective treatments (METs)
- Comment #2: For case 2, line 115 - you state that the patient could not run but only had an EDSS of 2. Could you clarify if there was a gait/walking problem and how that would only lead to an EDSS of 2?
Answer: it has been clarified that the patient couldn’t run properly, but his gait was normal and the EDSS was 2.0.
- Comment #3: line 126 – it states that there were significant relapses but no new MRI lesions. Could you clarify if there were really no new MRI lesions at the time of the relapse?
Answer: it has been clarified that:
- brain and spinal cord MRIs has been performed 3 and 6 months after the relapse and then yearly up to three years
- they didn’t show any new or enlarging or enhancing lesion.
- Comment #4: line 153 - Please clarify the single intrathecal oligoclonal band and several bands in both plasma and CSF?
Answer: it has been clarified that CSF examination revealed several oligoclonal bands in both serum and CSF and an isolated monoclonal band only in CSF.
- Comment #5: line 187 - you state that the 2017 criteria were the first to apply to children but the 2011 criteria did include children. The 2017 paper states that the McDonald criteria (including MRI and CSF) were most applicable to children age 11 and up.
Answer: it has been revised.
Round 2
Reviewer 1 Report
The quality of scientific contents and language of this case series remain suboptimal after revision. The manuscript has not been well prepared, as indicated by the multiplicity of problems listed below. In addition, the authors did not provide the "Response to Reviewer" Letter. 1. According to Table 2, case 3 and case 4 had their disease onset and first disease-modifying drug use at the same age. If this is true, then how did the authors obtain the annualized relapse rate for these patients? 2. in case 1, case 2, and case 5, more than one kind of high-efficacy therapy (HET) have been used, apparently because some HET did not achieve adequate control of disease activity. I would suggest that the authors also calculate the annualized relapse rate for the duration of each individual HET, which may serve as a basis for comparisons across different medications. 3. It is stated that extended interval dosing of Natalizumab was used in some cases with the intention to reduce PML risk. The authors may need to justify if this regimen is still qualitied as a HET. 4. In case 2, after failure of natalizumab, what was the consideration in choosing to switch to alemtuzumab (off label use in Italy, according to the authors) rather than fingolimod ? Minor issues: 1. In the first paragraph of Introduction, "AOMS" should be defined at its first appearance. 2. In 3.1.2 Case 2, ".... The diagnosis of POMS was made on the basis of his clinical data." This statement is quite vague and provides no meaningful information to the readers. 3. In 3.1.2 Case 2, "anti-AQ4" is not a formal abbreviation. 4. In 3.1.2 Case 2, Internuclear ophthalmoplegia (INO) -> already defined in Case 1 5. In 3.1.3 Case 3, "Anti-AQ 4" 6. In Materials and Methods, ".... according to He et al [6]." Who is "He" ??Author Response
We thank the reviewers for their comments, which have allowed us to improve out paper.
We hope that it will now be deemed fit for publication in Children.
Please find below a point-by-point answer to the reviewers’ comments.
Reviewer #1
Major Comments
- Comment #1: " According to Table 2, case 3 and case 4 had their disease onset and first disease-modifying drug use at the same age. If this is true, then how did the authors obtain the annualized relapse rate for these patients?”.
Answer: Annualized relapse rate (ARR) has been calculated as the number of relapses with onset occurring during the period of time between disease onset and first DMD, adjusted to a one-year period. Case 3 had 2 relapses in 8.5 months, resulting in a 2.8 ARR. Case 4 had MS diagnosis at the age of 15, she reported a relapse (two episodes in a month characterized by bilateral paresthesias in the legs) when she was fourteen and she started the first DMD 23 months after disease onset, resulting in a 1.0 ARR; Case 3 age of onset has been corrected to 14 in the table. We apologize for the error.
- Comment #2: “in case 1, case 2, and case 5, more than one kind of high-efficacy therapy (HET) have been used, apparently because some HET did not achieve adequate control of disease activity. I would suggest that the authors also calculate the annualized relapse rate for the duration of each individual HET, which may serve as a basis for comparisons across different medications.
Answer: Table two has been updated to include DMDs sequencing and ARR on each treatment
- Comment #3: “It is stated that extended interval dosing of Natalizumab was used in some cases with the intention to reduce PML risk. The authors may need to justify if this regimen is still qualitied as a HET.”
Answer: It has been clarified that Extended Interval Dosing (EID) Natalizumab efficacy was similar to standard dosing according to the results of NOVA clinical trial, therefore EID Natalizumab is still considered a HET.
- Comment #4: “In case 2, after failure of natalizumab, what was the consideration in choosing to switch to alemtuzumab (off label use in Italy, according to the authors) rather than fingolimod?”
Answer: In the discussion, it has been clarified that we escalated to Alemtuzumab for efficacy reason: natalizumab showed a greater efficacy than fingolimod in the REVEAL study, so we didn’t use a potentially less effective drug.
Minor Comments
- Comment # 1: In the first paragraph of Introduction, "AOMS" should be defined at its first appearance.
Answer: it has been corrected
- Comment #2: In 3.1.2 Case 2, ".... The diagnosis of POMS was made on the basis of his clinical data." This statement is quite vague and provides no meaningful information to the readers. .
Answer: It has been clarified that the diagnosis of dissemination in time and space was made on clinical basis (two relapses in two different functional system), according to McDonald revised criteria.
- Comment #3: In 3.1.2 Case 2, "anti-AQ4" is not a formal abbreviation.
Answer: it has been corrected to the formal abbreviation anti-AQP4
- Comment #4: In 3.1.2 Case 2, Internuclear ophthalmoplegia (INO) -> already defined in Case 1.
Answer: it has been corrected
- Comment #5: In 3.1.3 Case 3, "Anti-AQ 4"
Answer: it has been corrected to the formal abbreviation anti-AQP4
- Comment #6: In Materials and Methods, ".... according to He et al [6]." Who is "He" ??.
Answer: the reference has been corrected; we apologize for the error.

Round 3
Reviewer 1 Report
I appreciate the efforts made by the authors in addressing the questions I raised in previous rounds of review. The case series appears to be more coherent now. Still, there are some issues that require further clarification. 1.(Abstract) "Materials and Methods: 2 out of 5 consecutive POMS were administered HET as initial therapy after diagnosis." I think this statement is not consistent with the descriptions in the manuscript. 2. The authors add a reference (Ref. 22) to support their use of extended-interval dosing of natalizumab. However, in the cited study, natalizumab extended-interval dosing is defined as once every 6 weeks "following at least 1 year of once every 4 weeks dosing". Therefore, it is questionable if earlier initiation of extended-interval dosing could still be viewed as high efficacy treatment for MS. The authors of the cited article (Foley et al.) also explain the reason why initiation of natalizumab treatment with extended-interval dosing might result in inadequate protection by referring to the article: KK Muralidharan, D Steiner, D Amarante, et al. Exposure–disease response analysis of natalizumab in subjects with multiple sclerosis. J Pharmacokinet Pharmacodyn, 44 (2017), pp. 263-275. 3. (Discussion, page 7) The authors add a paragraph with regard to cognitive assessment in these patients. I think it would be more appropriate to move this part to "Materials and Methods" and "Results". 4. Proofreading and editing of the manuscript are necessary. To name but a few, 4-1. Both DMD and DMT are used in the text to refer to disease-modifying therapies, and the abbreviation "DMD" is defined differently in the Introduction and in the footnote of Table 2. Please rectify to make them consistent throughout the text. 4-2. Some capital letter is used inappropriately. For example, (Introduction) Adult-Onset Multiple Sclerosis ....; (3.1.2) Pontine lesion .... 4-3. (Abstract) "Materials and Methods: .... consecutive POMS were administered HET as initial therapy after diagnosis." Would it be better to say ".... consecutive patients with POMS were ...." ? 4-4. (page 5) "3.2. Figures, Tables and Schemes" ??? 4-5. (page 7) ".... no-responders to ...." -> should be "non-responders"Author Response
We thank the reviewers for their comments, which have allowed us to improve out paper.
We hope that it will now be deemed fit for publication in Children.
Please find below a point-by-point answer to the reviewer’s comments.
Reviewer #1
Comments
- Comment #1: " (Abstract) "Materials and Methods: 2 out of 5 consecutive POMS were administered HET as initial therapy after diagnosis." I think this statement is not consistent with the descriptions in the manuscript.”
Answer: This statement has been corrected
- Comment #2: “The authors add a reference (Ref. 22) to support their use of extended-interval dosing of natalizumab. However, in the cited study, natalizumab extended-interval dosing is defined as once every 6 weeks "following at least 1 year of once every 4 weeks dosing". Therefore, it is questionable if earlier initiation of extended-interval dosing could still be viewed as high efficacy treatment for MS. The authors of the cited article (Foley et al.) also explain the reason why initiation of natalizumab treatment with extended-interval dosing might result in inadequate protection by referring to the article: KK Muralidharan, D Steiner, D Amarante, et al. Exposure–disease response analysis of natalizumab in subjects with multiple sclerosis. J Pharmacokinet Pharmacodyn, 44 (2017), pp. 263-275.”
Answer: In the discussion it has been clarified that extended interval dosing natalizumab was started in case 4 after one year of standard dosing, but in case 3 it was started after six months of standard dosing, and it is questionable if earlier initiation of extended interval dosing could still be considered a highly effective treatment.
- Comment #3: “(Discussion, page 7) The authors add a paragraph with regard to cognitive assessment in these patients. I think it would be more appropriate to move this part to "Materials and Methods" and "Results".
Answer: The paragraph regarding cognitive assessment has been moved from “Discussion” to “Materials and Methods” and “Results”.
- Comment #4: “Proofreading and editing of the manuscript are necessary. To name but a few, 4-1. Both DMD and DMT are used in the text to refer to disease-modifying therapies, and the abbreviation "DMD" is defined differently in the Introduction and in the footnote of Table 2. Please rectify to make them consistent throughout the text. 4-2. Some capital letter is used inappropriately. For example, (Introduction) Adult-Onset Multiple Sclerosis ....; (3.1.2) Pontine lesion .... 4-3. (Abstract) "Materials and Methods: .... consecutive POMS were administered HET as initial therapy after diagnosis." Would it be better to say ".... consecutive patients with POMS were ...." ? 4-4. (page 5) "3.2. Figures, Tables and Schemes" ??? 4-5. (page 7) ".... no-responders to ...." -> should be "non-responders"
Answer: Proofreading and editing of the manuscript have been done
